# Equilibrium Refinement for the Age of Machines: The One-Sided Quasi-Perfect Equilibrium

**Gabriele Farina**
Computer Science Department
Carnegie Mellon University
gfarina@cs.cmu.edu

**Tuomas Sandholm**
Computer Science Department, CMU
Strategy Robot, Inc.
Strategic Machine, Inc.
Optimized Markets, Inc.
sandholm@cs.cmu.edu

## Abstract

In two-player zero-sum extensive-form games, Nash equilibrium prescribes optimal strategies against perfectly rational opponents. However, it does not guarantee rational play in parts of the game tree that can only be reached by the players making mistakes. This can be problematic when operationalizing equilibria in the real world among imperfect players. Trembling-hand refinements are a sound remedy to this issue, and are subsets of Nash equilibria that are designed to handle the possibility that any of the players may make mistakes. In this paper, we initiate the study of equilibrium refinements for settings where one of the players is perfectly rational (the "machine") and the other may make mistakes. As we show, this endeavor has many pitfalls: many intuitively appealing approaches to refinement fail in various ways. On the positive side, we introduce a modification of the classical quasi-perfect equilibrium (QPE) refinement, which we call the *one-sided quasi-perfect equilibrium*. Unlike QPE, one-sided QPE only accounts for mistakes from one player and assumes that no mistakes will be made by the machine. We present experiments on standard benchmark games and an endgame from the famous man-machine match where the AI *Libratus* was the first to beat top human specialist professionals in heads-up no-limit Texas hold'em poker. We show that one-sided QPE can be computed more efficiently than all known prior refinements, paving the way to wider adoption of Nash equilibrium refinements in settings with perfectly rational machines (or humans perfectly actuating machine-generated strategies) that interact with players prone to mistakes. We also show that one-sided QPE tends to play better than a Nash equilibrium strategy against imperfect opponents.

## 1 Introduction

The Nash equilibrium solution concept prescribes optimal strategies against perfectly rational opponents. However, it is well known that it has serious shortcomings when used to prescribe strategies to be deployed against imperfect opponents who may make mistakes. Even in two-player zero-sum games, it does not guarantee rational play in parts of the game tree that can only be reached if the players make mistakes. As a very simple perfect-information-game example, consider the game in Figure 1 (Left). The bold lines show one of the Nash equilibria of the game. It does not matter whether the white player acting at $B$ chooses move $l$ or move $r$ because he never gets to move if the black player acting at $A$ plays rationally. So, in Nash equilibrium, the white player can choose move $l$. However, if the black player makes a mistake and chooses move $b$, it would be better for the white player to choose move $r$ (thus getting a payoff of $5$ instead of $0$). So, in that part of the game where the black player has made the mistake, the white player's Nash equilibrium strategy is not rational.

35th Conference on Neural Information Processing Systems (NeurIPS 2021).

In game-theoretic terms, it is not *sequentially rational*. This is problematic when operationalizing

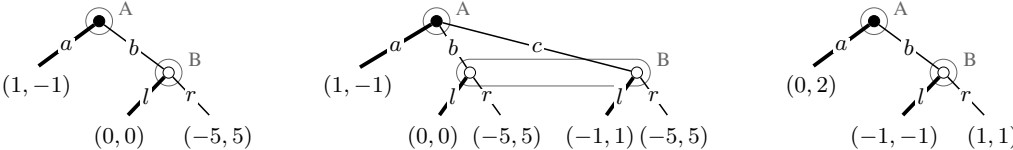

Figure 1: Small extensive-form games in which a non-sequentially-rational Nash equilibrium is highlighted. The first payoff of each outcome is assigned to the black player, the second to the white player.

equilibria in the real world among imperfect players. While in the particular example of Figure 1 (Left) the issue could be resolved by using an equilibrium refinement called *subgame perfect* Nash equilibrium, that solution concept does not refine solutions much in imperfect-information games, where few subgames (nodes of the game tree that are alone in their information set) exist. For example, consider the example in Figure 1 (Center): the white player acting at information set $B$ does not have any subgame, and therefore the highlighted sequentially-irrational Nash equilibrium is subgame perfect. Another refinement of Nash equilibrium is *undominated Nash equilibrium (UNE)*, that is, Nash equilibrium where the pure strategies in the support of the equilibrium do not include strongly dominated strategies. UNE would remove the unreasonable Nash equilibria in the games of Figure 1 (Left and Center), but there are other games where UNEs can be sequentially irrational [21]. In general, undomination and sequential rationality are incomparable in the sense that neither implies the other [21].

For imperfect-information games, the main family of equilibrium refinements (for example against sequential irrationality) is *trembling-hand refinements*, which are significantly more intricate than subgame perfection [28, 24, 31]. Trembling-hand equilibria are a subset of Nash equilibria that are designed to handle the possibility that any of the players may make mistakes. Roughly speaking, each player is assumed to make every mistake with some small probability, and trembling-hand equilibria are the limit points of the sequence that arises as that trembling (that is, mistake) probability approaches zero. As we will summarize later, there are multiple trembling-hand refinements that differ based on how the trembling constraints are set up.

In this paper, we initiate the study of equilibrium refinements for settings where one of the players is perfectly rational (the "machine") and only the other may make mistakes. We will conventionally refer to the latter player as the "imperfect" player. This is a setting that is becoming increasingly common in AI applications such as recreational games, military settings, and business [23, 1, 2, 4]. As we show, this endeavor has many pitfalls: intuitively appealing approaches to refinement fail in various ways. On the positive side, we introduce a modification of the classical quasi-perfect equilibrium (QPE) refinement, which we call the *one-sided quasi-perfect equilibrium*. Unlike QPE, one-sided QPE only accounts for mistakes from the imperfect player and assumes that no mistakes will be made by the machine. We present extensive experiments on standard benchmark games and an endgame from the famous man-machine match where the AI *Libratus* was the first to beat top human specialist professionals in heads-up no-limit Texas hold'em poker. We show that one-sided QPE can be computed more efficiently than any prior trembling-hand refinements, paving the way to wider adoption of Nash equilibrium refinements in settings with perfectly rational machines (or humans perfectly actuating strategies) that interact with imperfect players prone to mistakes. We also show that one-sided QPE tends to play better than a Nash equilibrium strategy against imperfect opponents.

## 2 Extensive-form games

In this section we review standard concepts in the theory of extensive-form (that is, tree-form) games. We will focus on two-player games with perfect recall and (potentially) imperfect information. An extensive-form game is a game played on a finite tree with payoffs at the leaves. Each node in the tree belongs to exactly one of the two players (which we call as Player 1 and Player 2), or belongs to a fictitious third player—called the *nature* player—whose actions are sampled from a known distribution. We will sometimes denote the opponent of Player $i \in \{1, 2\}$ with the symbol $-i$. The set of nodes that belong to the same player $i \in \{1, 2\}$ is split into a partition $\mathcal{I}_i$, called the *information*

*partition* of Player $i$. Each set $I$ in the partition is called an *information* set: two nodes belong to the same information set when the player cannot distinguish between them when he or she needs to act at them. When all information sets for all players are singleton sets, the player has no uncertainty about where in the game tree the are; in that case, the game is said to have perfect information. In this paper we only consider *perfect-recall* games. This means that the information partition of both players is such that any two nodes in the same information set share the same sequence of actions of that player from the root to those nodes. Intuitively, this means that the players do not forget about their past actions and observations in the game. Given two information sets $I', I \in \mathcal{I}_i$ for the same player $i$, we say that $I'$ is a successor of $I$, written $I' \succeq I$, if the sequence of actions of Player $i$ on the path from the root to any node in $I$ passes through some node in $I'$. Let $A(I)$ denote the set of actions available at any of the nodes in the information set $I \in \mathcal{I}_i$. The set of *sequences* for Player $i$ is defined as the set of all information set-action pairs $\sigma \in \{(I, a) : I \in \mathcal{I}_i, a \in A(I)\}$. Each sequence $\sigma = (I, a)$ uniquely identifies a path from the root of the game tree down to action $a$ and information set $I$. The length $|\sigma|$ is defined as the number of Player $i$'s actions on that path. Conceptually, a strategy in an extensive form games specifies a probability distribution over the set of actions $A(I)$ available at each information set $I \in \mathcal{I}_i$. We will represent strategies as vectors using the *sequence-form representation* [15, 32, 26]. In that representation, the vector corresponding to a strategy has one coordinate per each sequence of the player, indicating the product of the probabilities of the player's actions in that sequence. It is well-known that under that representation, the set of all well-formed sequence-form strategies for the player is a convex polytope $\boldsymbol{F}_i \boldsymbol{x} = \boldsymbol{f}_i$ for a suitable pair of sparse matrix and vector $\boldsymbol{F}_i, \boldsymbol{f}_i$ encoding probability-mass-conservation constraints (called sequence-form constraints). A strategy $\boldsymbol{\pi}_i$ for Player $i$ is a *best response* to a given strategy $\boldsymbol{\pi}_{-i}$ of the opponent if no other strategy for Player $i$ gives to Player $i$ strictly greater expected utility against $\boldsymbol{\pi}_{-i}$.

## 3 Nash equilibrium and its refinements

Nash equilibrium is the most widely used solution concept in game theory. A pair of strategies $(\boldsymbol{x}_1, \boldsymbol{x}_2)$ for two players in a game is a Nash equilibrium if neither player is (strictly) better off by deviating to any other strategy if the opponent does not deviate. In the special case of *zero-sum* games, it is a celebrated result that the set of Nash equilibria $(\boldsymbol{x}_1, \boldsymbol{x}_2)$ is the set of solutions to the bilinear saddle point optimization problem

$$\max_{\substack{\boldsymbol{F}_1 \boldsymbol{x}_1 = \boldsymbol{f}_1 \\ \boldsymbol{x}_1 \geq \boldsymbol{0}}} \min_{\substack{\boldsymbol{F}_2 \boldsymbol{x}_2 = \boldsymbol{f}_2 \\ \boldsymbol{x}_2 \geq \boldsymbol{0}}} \boldsymbol{x}_1^\top \boldsymbol{A}_1 \boldsymbol{x}_2.$$

In the rest of the paper we will focus on zero-sum games. There, any strategy that is part of a Nash equilibrium is an *optimal* strategy against any Nash equilibrium strategy of the opponent, that is, against any rational opponent. Furthermore, if the opponent plays any strategy other than an equilibrium strategy, that can only increase our expected utility.

However, as we illustrated in Figure 1, Nash equilibrium suffers from the severe issue of being unable to capitalize on opponent mistakes when the opponent is, in fact, not perfectly rational. This is true already in the zero-sum game setting, which is the focus of this paper.[1] While this issue is easy to avoid in perfect-information games by restricting attention to subgame-perfect Nash equilibria, the imperfect-information case has been significantly more nuanced historically. The introduction of *sequential rationality* was a seminal step down that avenue [16].

We devote the rest of this section to the standard solution concepts that guarantee sequential rationality (thereby soundly remedying the shortcomings of Nash equilibrium), that is, *trembling-hand equilibrium refinements*. The fundamental idea behind trembling-hand equilibria is to modify the Nash equilibrium optimization problem by adding constraints that force lower bounds of some forms on all action probabilities so as to force all parts of the game tree to be taken into consideration. A trembling-hand equilibrium is then a limit point of those constrained Nash equilibria as the lower bounds approach zero. Two different classes of trembling-hand refinements are known, and they differ in the way they force the lower bounds. The *extensive-form perfect equilibrium* concept (Section 3.1) enforces that each action be picked with at least some probability. That is, there is a uniform lower

---

[1]In non-zero-sum games, this issue is only exacerbated further. For example, non-credible threats can be supported in Nash equilibrium. See also Section 4.3.

bound on all action probabilities. The *quasi-perfect equilibrium* concept (Section 3.2) changes this by requiring lower bounds on *sequences* of actions rather than individual actions.

**3.1  Extensive-form perfect equilibrium.**  Given a game $\Gamma$, the idea behind extensive-form perfect equilibria (EFPEs) is to introduce a parameter $\epsilon > 0$ (the *trembling magnitude*), and consider the *perturbed game* $\Gamma(\epsilon)$ in which each player can only play strategies that put probability mass $\geq \epsilon$ on every action. An EFPE is then any limit point of Nash equilibria for the games $\Gamma(\epsilon)$ as $\epsilon \to 0^+$ [28]. It is well-known (e.g., Kreps and Wilson [16]) that every game has at least one EFPE, and that EFPEs are sequentially rational.

**3.2  Quasi-perfect equilibrium.**  Quasi-perfection, introduced by van Damme [30], is significantly more intricate to define than extensive-form perfection. Instead of giving an explicit lower bound on the probability with which each action needs to be selected, the definition of a quasi-perfect equilibrium (QPE) relies on a refined notion of best response. We now give one of the multiple known equivalent definitions, and we present it for the special case of two-player games only. Several equivalent definitions that apply to more general games can be found in the original work by van Damme, as well as in the work by Miltersen and Sørensen [22] and Gatti et al. [11].

**Definition 1** (*$I$-local purification*).  *Let $i \in \{1, 2\}$ be a player, $\boldsymbol{\pi}$ be a strategy for Player $i$, and let $I \in \mathcal{I}_i$ be an information set. We say that a strategy $\boldsymbol{\pi}'$ for Player $i$ is an $I$-local purification of $\boldsymbol{\pi}$ if $\boldsymbol{\pi}'$ is deterministic at any information set $I' \succeq I$, and coincides with $\boldsymbol{\pi}$ at any other information set. When $\boldsymbol{\pi}'$ is an $I$-local purification of $\boldsymbol{\pi}$, we further say that*

- *$\boldsymbol{\pi}'$ is $\epsilon$-consistent with $\boldsymbol{\pi}$ if, for all $I' \succeq I$, $\boldsymbol{\pi}'$ assigns probability $1$ only to actions that have probability $\geq \epsilon$ in $\boldsymbol{\pi}$;*
- *$\boldsymbol{\pi}'$ is optimal against a given strategy of the opponent if no other $I$-local purification of $\boldsymbol{\pi}$ achieves (strictly) higher expected utility against said strategy of the opponent.*

**Definition 2** (*$\epsilon$-quasi-perfect best response*).  *A strategy $\boldsymbol{\pi}_i$ is an $\epsilon$-quasi-perfect best response to the opponent strategy $\boldsymbol{\pi}_{-i}$ if (i) $\boldsymbol{\pi}$ assigns strictly positive probability to all actions of Player $i$; and (ii) for all information sets $I \in \mathcal{I}_i$ of Player $i$, every $\epsilon$-consistent $I$-local purifications of $\boldsymbol{\pi}_i$ (Definition 1) is optimal for $\boldsymbol{\pi}_{-i}$. A strategy profile $(\boldsymbol{\pi}_1, \boldsymbol{\pi}_2)$ where each strategy is an $\epsilon$-quasi-perfect best response to the opponent's strategy is called an $\epsilon$-quasi-perfect strategy profile.*

**Definition 3** (*Quasi-perfect equilibrium*).  *A quasi-perfect equilibrium is any limit point of $\epsilon$-quasi-perfect strategy profiles as $\epsilon \to 0^+$.*

It is known since the work by Miltersen and Sørensen [22] that some QPEs (we call them *Miltersen-Sorensen QPEs*) can be computed in any two-player game as the limit point of Nash equilibria of perturbed games $\Gamma(\epsilon)$, akin to EFPE. The subtlety is that while in EFPE each perturbed game $\Gamma(\epsilon)$ mandates a lower bound of $\epsilon$ on the probability of playing each action, in the case of a Miltersen-Sorensen QPE the lower bounds are given on the probability of each *sequence* of actions. Specifically, for any $\epsilon > 0$ and for each player $i \in \{1, 2\}$, let $\boldsymbol{\ell}_i : \epsilon \to \mathbb{R}_{>0}^{|\Sigma_i|}$ denote the vector parametrized on $\epsilon$ and indexed on the sequences $\Sigma_i$ of Player $i$, whose entries are defined as

$$\ell_i(\epsilon)[\sigma] = \epsilon^{|\sigma|} \qquad \forall \sigma \in \Sigma_i, \tag{1}$$

where $|\sigma|$ denotes the number of actions for Player $i$ in the sequence $\sigma$. Miltersen and Sørensen [22] prove that any limit point of the solution to the perturbed optimization problem

$$\max_{\substack{\boldsymbol{F}_1 \boldsymbol{x}_1 = \boldsymbol{f}_1 \\ \boldsymbol{x}_1 \geq \boldsymbol{\ell}_1(\epsilon)}} \min_{\substack{\boldsymbol{F}_2 \boldsymbol{x}_2 = \boldsymbol{f}_2 \\ \boldsymbol{x}_2 \geq \boldsymbol{\ell}_2(\epsilon)}} \boldsymbol{x}_1^\top \boldsymbol{A}_1 \boldsymbol{x}_2. \tag{2}$$

is a (Miltersen-Sorensen) QPE.[2]

**3.3  A word of caution: Not all natural vanishing perturbations lead to sequential rationality**

We found a potential pitfall when introducing lower bounds on sequence probabilities with the hope of computing a trembling-hand refinement. Not all vanishing perturbations $\boldsymbol{\ell}_1(\epsilon), \boldsymbol{\ell}_2(\epsilon)$ in the QPE formulation (2) lead to a sequentially-rational equilibrium. For example, it is natural to wonder

---

[2]Recently, Gatti et al. [11] took this construction further, and showed that *any* QPE can be expressed as a limit point of solutions to (2), as long as more general vectors of polynomials $\boldsymbol{\ell}_1, \boldsymbol{\ell}_2$ are used than in (1). In this paper we will focus on Miltersen-Sorensen-style perturbation as defined in (1).

whether it is *really* necessary to consider lower bounds of the form $\epsilon^{|\sigma|}$ instead of, for example, the uniform lower bound $\epsilon$ for all sequences. After all, surely a uniform lower bound of $\epsilon$ would still force the whole game to be explored, wouldn't it? While appealing on the surface, such a uniform lower bound might result in a solution that is not even subgame perfect, much less sequentially rational! In particular, consider the perfect-recall game in Figure 2. We prove in the appendix that for any choice of $\epsilon \in [0, 1/4]$, the only Nash equilibrium of the perturbed game assigns probability $1 - \epsilon$ to action $r$ of Player 2, and probability $1/2$ to actions $c$ and $d$ of Player 1. So, as $\epsilon \to 0^+$, any limit point sees Player 2 pick action $r$ with probability 1 and Player 1 randomizing uniformly between actions $c$ and $d$, despite action $d$ being strictly dominated. Thus, both players will act irrationally (with Player 1 not even playing a best response in the subtree rooted at C) should Player 1 make the mistake of picking action $b$ instead of $a$ at the root A. The resulting equilibrium is not subgame perfect, and consequently it cannot be sequentially rational [16, Proposition 3].

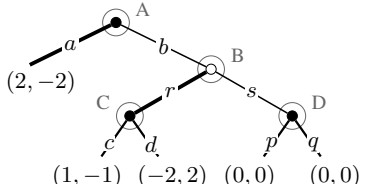

| Action | Probability |
| --- | --- |
| $a$ | $1 - 4\epsilon$ |
| $b$ | $4\epsilon$ |
| $c, d, p, q$ | $1/2$ |
| $r$ | $1 - \epsilon$ |
| $s$ | $\epsilon$ |

Figure 2: Small perfect-information game that illustrates that uniform $\epsilon$ lower bounds can induce irrational behavior. Black nodes belong to Player 1, the white node belongs to Player 2.

## 4  One-sided quasi-perfect equilibrium

All of the trembling-hand equilibrium refinements summarized in Section 3 are *two sided*, in the sense that both players are trembling. That two-sidedness comes at a computational cost: both the domain of the maximization and minimization problem in the saddle point formulations (for example, Equation (2) in the case of QPE) of the refinements are perturbed, making the computation of a limit point computationally expensive. Yet, in many strategic interactions of interest, a player might be concerned about being able to capitalize on the opponent's mistakes, but not about making mistakes of her own. After all, in the age of machines, that player might well be a bot interacting strategically (for example, playing a poker tournament) against imperfect opponents. In that situation, the player in question might therefore seek, in the interest of lowering the computational requirement of computing a robust strategy, to find equilibrium points that are robust to perturbations *of the opponent's strategy only*, thereby breaking the two-sidedness of all known trembling-hand equilibrium refinements.

In this paper, we introduce the first *one-sided* trembling-hand refinement, which we coin *one-sided quasi-perfect equilibrium*. Because of the asymmetric role of the players, from now on we stop referring to the players as Player 1 and 2, and adopt the terms *machine player* and *imperfect player* to highlight their asymmetric role. The machine player is assumed to never make mistakes: lower bounds on the probability of play (the "trembling hands") are only introduced for the imperfect player. Accordingly, from now on we will drop subscripts 1 and 2 to denote quantities that belong to the players, and will use $m$ and $h$ for quantities belonging to the machine and the imperfect player, respectively.

**4.1  Definition and preliminary considerations.** In order to formally define the one-sided quasi-perfect equilibrium solution concept, we start by removing some of the symmetry between the two players in Definition 2. In particular, we introduce the following notion.

**Definition 4** (One-sided quasi-perfect equilibrium). *We call a strategy profile $(\boldsymbol{\pi}_m, \boldsymbol{\pi}_h)$ a one-sided $\epsilon$-quasi-perfect strategy profile if $\boldsymbol{\pi}_h$ is an $\epsilon$-quasi-perfect best response (Definition 2) to $\boldsymbol{\pi}_m$, and $\boldsymbol{\pi}_m$ is a best response to $\boldsymbol{\pi}_h$. We say that $(\boldsymbol{\pi}_m, \boldsymbol{\pi}_h)$ is an one-sided quasi-perfect equilibrium if it is the limit point of one-sided $\epsilon$-quasi-perfect strategy profiles, as $\epsilon \to 0^+$.*

One-sided quasi-perfect equilibria do indeed form a refinement of the Nash equilibrium, as we establish in the next theorem (see the appendix for a proof).

**Theorem 1.** *Every one-sided quasi-perfect equilibrium is a Nash equilibrium.*

At this stage it is still technically unclear whether one-sided quasi-perfect equilibria exist at all. To show existence, as a first step we slightly extend the result by Miltersen and Sørensen [22] that we mentioned in Section 3.2, and show that one-sided $\epsilon$-quasi-perfect strategies exist, and that one can be computed as the solution to a bilinear saddle point problem.

**Lemma 1.** *Consider the bilinear saddle point problem*

$$\max_{\substack{\boldsymbol{F}_m \boldsymbol{x}_m = \boldsymbol{f}_m \\ \boldsymbol{x}_m \geq \boldsymbol{0}}} \min_{\substack{\boldsymbol{F}_h \boldsymbol{x}_h = \boldsymbol{f}_h \\ \boldsymbol{x}_h \geq \boldsymbol{\ell}_h(\epsilon)}} \boldsymbol{x}_m^\top \boldsymbol{A}_m \boldsymbol{x}_h \tag{3}$$

*where $\boldsymbol{\ell}_h(\epsilon)$ is as in Equation* (1)*. Then, for any $\epsilon > 0$ for which the domain of the minimization problem is nonempty, any solution to* (3) *is a one-sided $\epsilon$-quasi-perfect strategy profile.*

From here, the existence of one-sided quasi-perfect equilibria can be established with a straightforward compactness argument. The domain of the minimization problem of (3) becomes nonempty for small enough values of the trembling magnitude $\epsilon > 0$. Therefore, for small enough $\epsilon$ the domains of the maximization and minimization problem in (3) are compact and nonempty. That, combined with the fact that the objective function is bilinear, immediately guarantees that (3) admits a solution for any small enough $\epsilon > 0$. Furthermore, such a solution belongs to the Cartesian product of the two players' sequence-form polytopes—a compact set—thereby guaranteeing that a limit point as $\epsilon \to 0^+$ exists as a valid strategy profile. So, Lemma 1 immediately implies the following corollary.

**Corollary 1.** *Every two-player zero-sum extensive-form game with perfect recall has at least one one-sided quasi-perfect equilibrium.*

**4.2 One-sided QPEs as trembling linear programs.** In this subsection, we show that the problem of computing a one-sided quasi-perfect equilibrium strategy $\boldsymbol{x}_m$ for the machine player can be cast to a linear program parameterized by the trembling magnitude $\epsilon$. We call such a linear program a *trembling linear program*, in concordance with nomenclature in the prior work by Farina et al. [7]. Specifically, the following result, which follows by linear programming duality, will be central in our discussion. An elementary proof is offered in the appendix.

**Proposition 1.** *Any limit point of solutions to the trembling linear program*

$$\mathcal{P}(\epsilon) := \begin{cases} \arg\max_{\boldsymbol{x}_m} \ (\boldsymbol{A}_m \boldsymbol{\ell}_h(\epsilon))^\top \boldsymbol{x}_m + (\boldsymbol{f}_h - \boldsymbol{F}_h \boldsymbol{\ell}_h(\epsilon))^\top \boldsymbol{v} \\ \quad \text{s.t.} \ \ ① \ \boldsymbol{A}_m^\top \boldsymbol{x}_m - \boldsymbol{F}_h \boldsymbol{v} \geq \boldsymbol{0} \\ \qquad\quad ② \ \boldsymbol{F}_m \boldsymbol{x}_m \qquad\quad = \boldsymbol{f}_m \\ \qquad\quad ③ \ \boldsymbol{x}_m \geq \boldsymbol{0}, \ \ \boldsymbol{v} \ \text{free.} \end{cases}$$

*as the trembling magnitude $\epsilon \to 0^+$ is a one-sided quasi-perfect equilibrium strategy for the machine player.*

**4.3 A second word of caution: One-sided QPE is inappropriate for general-sum games.** Sequential irrationality occurs in both zero-sum and general-sum games because the Nash equilibrium concept does not consider mistakes by the players. In general-sum games, sequential irrationality is exacerbated further by the presence of non-credible threats (which cannot occur in zero-sum games because there are no actions that hurt both players). For example, consider the highlighted Nash equilibrium in the small general-sum game of Figure 1 (Right). If the black player acting at $A$ were to actually play $b$, it would be irrational for the white player to play $l$, which hurts both players. Effectively, the white player is "threatening" to play $l$ instead of $r$ to force the black player's hand and push the black player to settle for an inferior payoff of $0$. By forcing all players to account for mistakes, even their own, trembling-hand equilibria are able to prevent the irrationality stemming from non-credible threats. Indeed, when action $b$ is played with probability at least $\epsilon > 0$, no Nash equilibrium would support action $l$ for the white player. Hence, the equilibrium in Figure 1 (Right) cannot be an EFPE or a QPE, which are limit points of Nash equilibria of perturbed games. However, because in one-sided QPEs only trembles from one player are considered, the one-sided QPE concept is generally unable to prevent non-credible threats. Specifically, if the black player were the machine player, and the white player the imperfect player, the highlighted equilibrium in Figure 1 (Right) would be a one-sided QPE. This should serve as a cautionary tale against using one-sided QPE in general-sum settings.

# 5 Computation of one-sided QPEs

While QPEs and EFPEs can in theory be computed in polynomial time (in the size of the game) in two-player zero-sum games [22, 6], their computation is fraught with difficulties that have historically slowed down their adoption. First, trembling-hand refinements are *limit points*, and not just the solution to a numeric optimization problem. Second, finding a Nash equilibrium subject to trembling constraints (*i.e.*, constraints that enforce actions are played with given lower bounds on probability) becomes numerically unstable when $\epsilon$ is small. In this section we show that while the computation of one-sided QPEs is still a nontrivial task, one-sided QPEs have a nice property that makes their computation comparatively easier and more numerically stable than QPEs and EFPEs. Specifically, the trembling linear program Proposition 1 defined in Section 4.2 has the convenient, differentiating property that $\epsilon$ only appears in the objective function and *not* in the constraints. As summarized in the table, neither EFPEs nor (two-sided) QPEs enjoy this property. Indeed, the trembling LP formulation of EFPE needs to express the constraint that each action is picked with probability $\geq \epsilon$. That is expressed by constraints of the form $x_h[(I, a)] \geq \epsilon \cdot x_h[\sigma_h(I)]$ for all information sets $I$ and actions $a \in A_I$, and for that reason the known trembling LP formulations of EFPE have $\epsilon$ appear in the left-hand side

| Equilibrium | Depends on $\epsilon$? | | |
| | LHS | RHS | Obj |
| --- | --- | --- | --- |
| EFPE | ✓ | ✗ | ✗ |
| QPE | ✗ | ✓ | ✓ |
| One-sided QPE | ✗ | ✗ | ✓ |

(LHS) of the constraint matrix [6, 7]. On the other hand, the trembling LP formulation of regular, two-sided (Miltersen-Sorensen) QPE has a component-wise lower bound on the vector $\boldsymbol{x}_m$, *i.e.*, constraints of the form $\boldsymbol{x}_m \geq \boldsymbol{\ell}_m(\epsilon)$. So, in QPE $\epsilon$ appears also in the right-hand side (RHS) of the constraints [22]. Part of the high complexity in practice associated with the computation of EFPE and QPE is related to where $\epsilon$ appears. As a rule of thumb, having $\epsilon$ terms in the constraint (left-hand side) matrix makes the problem the hardest, as those terms impact the numerical stability of the basis matrix, which needs to be inverted (more precisely, factorized) after every pivoting step of the simplex algorithm. That is avoided in the somewhat easier case where $\epsilon$ only appears on the right-hand size of the constraints, though that case is still hard, given that the feasible set still depends on $\epsilon$, thereby making the task of maintaining feasibility as $\epsilon \to 0^+$ nontrivial. One-sided QPE avoids both of these issues, by only having a dependence on $\epsilon$ in the objective function: the feasible set of $\mathcal{P}(\epsilon)$ is constant, and only the coefficients of the objective function change (continuously) as $\epsilon \to 0^+$.

Currently, the only known algorithm for finding limit solutions to trembling linear programs (of which QPEs, one-sided QPEs, and EFPEs are examples) is via the algorithm of Farina et al. [7].[3] That algorithm computes a limit point of solutions of any trembling linear program, such as $\mathcal{P}(\epsilon)$ given in Section 4.2. At a high level it operates as follows. First, a value for $\epsilon^* > 0$ is chosen arbitrarily. Then, the linear program (LP) $\mathcal{P}(\epsilon^*)$ is solved numerically to optimality by using the simplex method, and an optimal basis for the LP is computed. A *basis-stability oracle* is then run, to check whether the basis that was computed numerically is *stable*, that is, whether it would remain optimal as $\epsilon \to 0^+$: if so, the algorithm terminates, otherwise the value of $\epsilon^*$ is reduced (typically by a multiplicative factor). The procedure is repeated for the new value of $\epsilon^*$, and so on. The loop continues until stability of the basis is established. While the details of the basis stability oracle are complex and beyond the scope of this paper, in the rest of the section we point out a few computational shortcuts that are enabled by the fact—discussed above—that the trembling linear program for one-sided QPE exhibits a dependence on $\epsilon$ only in the objective function. In the discussion, we assume some familiarity with the simplex algorithm and the concept of basic and non-basic columns.

- The feasible set in the trembling linear program for one-sided QPEs is independent of $\epsilon$. Hence, the optimal basis computed for the numeric perturbation value $\epsilon^*$ remains feasible even after the $\epsilon^*$ is reduced. So, at each iteration of the algorithm by Farina et al. [7], we can very effectively warm start the simplex method with the basis computed in the previous iteration, cutting through the first phase of the simplex algorithm (computation of a feasible basic solution), and jumping straight to pivoting until a new optimal basis is found. This shortcut is not possible in QPEs and EFPEs.

- Similarly, when evaluating whether a computed basis remains optimal when the limit is taken, we can soundly skip verifying feasibility in the limit: since the computed basis is optimal for a given

---

[3]Other algorithms provide approximate solutions to approximate solution concepts. For example, Farina et al. [8] and Kroer et al. [17] propose methods based on regret minimization and the excessive-gap technique [25], respectively, to approximate solutions to games subject to trembling constraints for a specific trembling magnitude. They do not provide any guarantee of actually finding (or even approximating) actual QPEs or EFPEs.

numeric value of $\epsilon^*$, it must be feasible. Because the feasible set does not change as $\epsilon$ goes to $0$, the basis must remain feasible in the limit. The same cannot be said for QPEs and EFPEs, for which instead it is necessary to investigate feasibility of the basis in the limit at every iteration.

- A consequence of the previous point is that only the reduced costs of the nonbasic columns matter when evaluating whether a given basis is optimal in the limit. Because $\epsilon$ does not appear in the constraint matrix that defines the trembling linear program for QPE and one-sided QPE, the reduced cost of every nonbasic column is a polynomial function of $\epsilon$, as opposed to a rational function like in the more general case. This greatly simplifies the implementation of the basis stability oracle for one-sided QPEs. Specifically, none of the discussion in the original paper by Farina et al. [7] about handling singular basis matrices and rational-function reduced costs using Laurent series applies to one-sided QPE. The same property applies to QPEs, but not to EFPEs. In the latter case, the stability oracle need to be implemented by taking into account the dependence of the constraint matrix on $\epsilon$.

In the experiments (Section 6), we implemented the algorithm by Farina et al. [7] by taking advantage of the computational shortcuts we just described. As we show empirically, the three considerations above translate into a reduced computational burden when computing one-sided QPEs compared to (regular, two-sided) QPEs and EFPEs.

## 6   Experimental evaluation

We compare one-sided QPEs against EFPE and (Miltersen-Sorensen) QPE, along two metrics: 1) the time required to compute the refinement, and 2) how the refinement fares against imperfect opponents, when compared to an exact but potentially unrefined Nash equilibrium computed by the two state-of-the-art linear programming solvers CPLEX and Gurobi. We implemented from scratch the algorithm by Farina et al. [7] to solve the trembling linear programs corresponding to the three equilibrium refinements. Our implementation takes the computational shortcuts described in Section 5 for one-sided QPEs (and for QPEs as well where applicable, that is, the third bullet point of that section). The algorithm is single-threaded, was implemented in C++, and was run on a machine with 32GB of RAM and an Intel processor running at a nominal speed of 2.4GHz per core.

As mentioned in Section 5, the algorithm computes, as an intermediate step at every iteration, an optimal basis of each trembling linear program where the perturbation magnitude $\epsilon$ has been set to a numerical value $\epsilon^*$. We start from the value $\epsilon^* = 10^{-6}$ and use Gurobi to solve the linear program. After the first iteration, if the basis is not stable, we re-solve the linear program, again for $\epsilon^* = 10^{-6}$ using Google's open-source linear programming solver (GLOP), which we modified so as to use 1000-digit precision floating point numbers via GNU's MPFR library. From there onward, after every unsuccessful iteration of the algorithm (that is, where the basis is not stable), the value of $\epsilon^*$ is decreased by a factor 1000 and solved again with our modified version of GLOP, until a stable basis is found. Unlike the original paper by Farina et al. [7], we do not employ a rational-precision implementation (that is, one that represents all numbers as ratios of integers to achieve an exact "infinite-precision" solution) of the simplex algorithm. Instead, we found our 1000-digit precision modified GLOP solver to be drastically faster, and we use it across the board in place of the rational simplex. The basis stability oracle is implemented using rational precision, as described in the original paper [7]. We use the GNU's GMP library to implement rational arithmetic. Therefore, our answer is exact (i.e., infinite-precision) even though the intermediate steps are not.

**6.1   Computation time.** We compare the compute time required to find a one-sided QPE strategy, (two-sided) QPE strategy, and EFPE strategy in six standard benchmark games: three instances of Leduc poker [29] of increasing size, one relatively large Goofspiel game [27], Liar's Dice, and one real river endgame from the "*Brains vs AI*" competition that was played by the *Libratus* AI. A description of each game is available in the appendix. To scale computation to the river endgame tractable, we solved the endgame using a coarser betting abstraction than the one used by Libratus. To our knowledge, it is the first time that sequentially-rational equilibria are investigated in real poker endgames. The dimensions of each game are listed in Table 1 (Left). Runtimes for each of the solution concepts are given in Table 1 (Right). We observe that one-sided QPE can be computed consistently faster (roughly by a factor 4-5x) than two-sided QPE, and the latter is usually twice as fast as EFPE. This is consistent with our discussion in Section 5. In the river endgame we implemented the sparsification technique described in Zhang and Sandholm [33] to bring down the number of nonzeros of the payoff matrix from 21 million to roughly 167 thousand combined nonzeros in the sparsification

when solving the linear program at each iteration (see the appendix for more details). In the river endgame, only one-sided QPE could be computed for both players. A (two-sided) QPE strategy could only be computed for Player 2, as Gurobi terminated abnormally due to numeric instability when solving for Player 1. None of the EFPE strategies could be computed, due to numerical instability in GLOP, which terminated with an error due to the basis being singular to working precision. With its 21 million terminal states, a one-sided QPE in the river endgame represents the upper limit of what equilibrium refinement technology can handle today. The numerical instability witnessed in QPE and EFPE for that benchmark game is well consistent with our discussion in Section 5.

| Game instance | Information sets | Sequences | Leaves | Player | One-sided QPE | QPE | EFPE |
|---|---|---|---|---|---|---|---|
| Leduc poker (5 ranks) | 780 | 1822 | 5500 | Player 1 | 222ms | 915ms | 1.70s |
| | | | | Player 2 | 387ms | 593ms | 1.11s |
| Leduc poker (9 ranks) | 2484 | 5798 | 32 724 | Player 1 | 1.58s | 5.81s | 12.70s |
| | | | | Player 2 | 2.65s | 6.73s | 14.95s |
| Leduc poker (13 ranks) | 5148 | 12 014 | 98 956 | Player 1 | 8.37s | 42.69s | 1m 36s |
| | | | | Player 2 | 17.35s | 38.94s | 1m 42s |
| Goofspiel (4 ranks) | 17 423 | 21 298 | 13 824 | Player 1 | 5.50s | 25.68s | 55.67s |
| | | | | Player 2 | 5.58s | 27.29s | 53.97s |
| Liar's dice | 24 576 | 49 142 | 147 420 | Player 1 | 26.86s | 2m 18s | 11m 02s |
| | | | | Player 2 | 28.36s | 2m 00s | 10m 25s |
| River endgame | 17 700 | 49 478 | 21 599 932 | Player 1 | 19m 36s | failure | failure |
| | | | | Player 2 | 10m 14s | 11m 52s | failure |

Table 1: (Left) Game instances we experiment on, and their size. (Right) Compute time necessary to find optimal strategies according to different solution concepts.

**6.2  Game-theoretic performance.** We compare the game-theoretic performance of the (refined) one-sided QPE strategies computed in Section 6.1, against the (unrefined) Nash equilibrium strategies computed by Gurobi and CPLEX, the two leading linear programming solvers. To do so, we generated a sequence of imperfect opponents by collecting the strategies output by CFR [34], a popular self-play algorithm that converges to Nash equilibrium in extensive-form games. We ran CFR for 10000 iterations. Then, we let our one-sided QPE strategies and the two unrefined Nash strategies (one from Gurobi and one from CPLEX) play against each of the imperfect opponents, and measured the difference in expected utility achieved by the strategies, normalized by the absolute value of the game. Results for the four largest games are shown in Figure 3 (plots for the two remaining games are in the appendix). For each game, the top plot shows the difference in expected utility (normalized by the absolute value of the game) obtained by our refined one-sided QPE strategy for machine Player 1 when compared to the Nash equilibrium strategy for Player 1 computed by Gurobi (solid blue line) and CPLEX (dashed orange line). The bottom plot shows the same, in the case where the machine player is set to Player 2 instead. The x-axis in each plot measures the exploitability of the imperfect player, normalized by the absolute value of the game. In the river endgame, the strategies computed by CPLEX are dual-infeasible (likely due to numeric instability), which explains why the curves do not pass through the origin. Our preliminary analysis suggests that refined strategies might indeed offer benefits over non-refined Nash equilibrium strategies. However, we point out that sometimes Gurobi and CPLEX happened to compute a strategy that was more exploitative than one-sided QPE for the specific irrationality of the CFR agents at that level of exploitability (for instance, in Liar's dice for exploitability up to $\approx 3$). This is consistent with the theory: nothing prevents Gurobi or CPLEX to terminate on a sequentially-rational strategies despite no constraints in that direction being imposed. As the experiments overall show, such an occurrence appears to be rare.

## 7  Conclusions, discussion, and future research

In this paper, we introduced a refined solution concept—the one-sided quasi-perfect equilibrium— suitable for zero-sum games where a "machine" player is not concerned by the possibility of making mistakes, but wants to make sure to account for the possible mistakes of an imperfect opponent. Along the way, we gave several fundamental results, and warned against common pitfalls. We showed that our refinement can be computed more effectively than the known existing alternatives in practice, and provided evidence that refined strategies might indeed outperform unrefined strategies, even

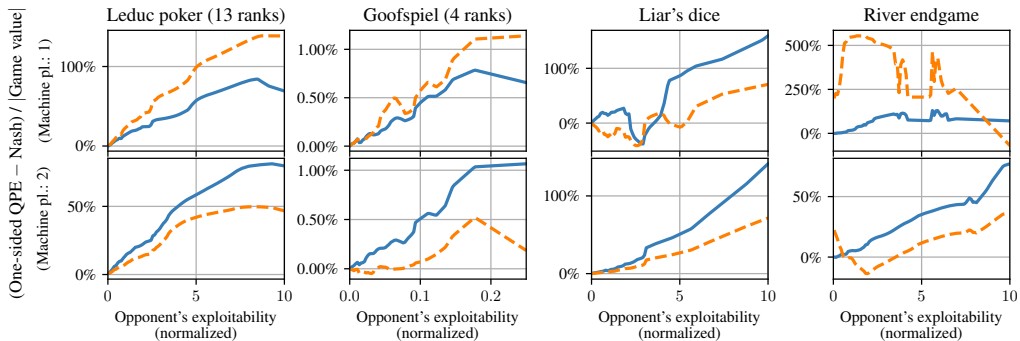

Figure 3: Increase in expected utility achieved by (refined) one-sided QPE strategies compared to the (unrefined) Nash equilibrium strategy for the game computed by Gurobi (solid blue line) and CPLEX (dashed orange line).

in benchmark games of interest including—for the first time—a real poker endgame. However, significantly more work needs to be done before refinements can be regarded as an appealing drop-in replacement of Nash equilibrium as a prescriptive tool. More work certainly remains to be done to enable computation of refined Nash equilibrium strategies remains a challenging problem (in our experiments, the computation of refined strategies was 1-2 orders of magnitude slower than the computation of unrefined strategies using commercial solvers).

Some readers might be wondering why we opted to only consider the one-sided version of quasi-perfect equilibrium, and not, say, also the one-sided version of extensive-form perfect equilibrium. The first reason stems from computational considerations: QPEs have the advantage over EFPEs that the trembling magnitude $\epsilon$ does not appear in the constraint matrix of the trembling linear program (see also Section 5), and that property carries over to their one-sided versions. The second reason is that there is consensus in the literature that QPEs are superior refinements than EFPEs [20, 13, 12]: (i) an EFPE may prescribe the players to play weakly dominated strategies, while a QPE never does; and (ii) in two-player games, a QPE is also a perfect equilibrium of the normal form, whereas EFPE is not. This led Mertens [20] to write: "*Observe that the "quasi-perfect" equilibria [..] are still sequential–and sequential equilibria have all backward-induction properties (e.g., Kohlberg and Mertens [14])–but are at the same time normal form perfect–which can be viewed as the strong version of undominated. (And every proper equilibrium is quasi-perfect.) Thus, by some irony of terminology, the "quasi"-concept seems in fact far superior to the original unqualified perfection itself.*". We leave the task of defining and exploring the theoretical and practical aspects of one-sided EFPEs and other one-sided equilibrium refinements as future research.

We remark that our one-sided QPE notion does not satisfy the traditional notion of sequential rationality, which is two-sided. In future work it might be interesting to define one-sided notions of sequential rationality and prove that our solution concept satisfies them. We observe that it is also possible to straightforwardly define one-sided notions of undominated Nash equilibrium. Among two-sided concepts, it has been shown experimentally that undominated equilibrium performs better than unrefined Nash equilibrium in reasonably-sized poker games [9], and even as well as trembling-hand refinements [5], at a fraction of the computational cost. However, the analysis was performed on small games only due to the scalability limitations of trembling-hand equilibrium finding at the time. The solution concept and algorithms introduced in this paper open the door for comparing one-sided QPE against one- and two-sided undominated Nash equilibrium at significantly larger scale.

Finally, we mention that empirically studying how much the theoretical benefit of trembling-hand solution concepts translates into practical performance against humans would be interesting. However, an appropriate analysis would require setting up human experiments, which is notoriously a complex undertaking. We leave that as a possible direction for future research. The analysis in our paper can be viewed as a first smoke test (in fact, the first of its kind, and on games significantly larger than anything prior refinement technology could scale to), but should not be taken as proof that refinements bring significant advantages compared to unrefined Nash strategies against human opponents. Our primary goal with this paper was to help scale up refinement computation technology to even start to enable those further experiments and investigations on such an important topic.

## Acknowledgments

This material is based on work supported by the National Science Foundation under grants IIS-1718457, IIS-1901403, and CCF-1733556, and the ARO under award W911NF2010081. Gabriele Farina is supported by a Facebook fellowship.

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
