## A  Equilibrium refinements and opponent exploitation

As mentioned in the introduction, Nash equilibrium refinements are designed to capitalize on opponent mistakes. They are a *passive* form of opponent exploitation. *Active* forms of opponent exploitation have been proposed (e.g., Ganzfried and Sandholm [10] and references therein), where typically the player is able to quantify the amount of value that the opponent is losing (compared to the value of the game, that is, the value obtained by fully rational players) due to mistakes, and use that as budget to more aggressively model and exploit the opponent. In other words, active opponent exploitation enables a learning agent to safely push themselves beyond a Nash equilibrium strategy to instead play an exploitative (but, in turn, exploitable—therefore, non-Nash) strategy against the opponent. That type of active exploitation is not possible with equilibrium refinements, which are Nash equilibria. However, one could imagine the two techniques working together: equilibrium refinements are a "free" avenue to capitalize on opponent mistakes, while guaranteeing no exploitability. Those opponent mistakes can than be used to control the risk exposure of active opponent exploitation techniques. This is another avenue of research that to our knowledge has not been explored so far.

## B  The example of Figure 2

Figure 2 is reproduced below for convenience. Let $0 \le \epsilon \le 1/4$. Action $a$ strictly dominates $b$, since all payoffs for the black player (Player 1) are strictly lower in the subtree rooted at $b$. Hence, the black player must minimize the probability mass put on the sequences that contain action $b$, compatibly with lower bounds. Because we are using uniform lower bounds $\epsilon$ on the probability of each sequence, the black player will need to put at least probability $\epsilon$ on the four sequences $bc, bd, bp, bq$. This can be achieved when $c, d, p, q$ are each selected with probability $1/2$ and action $b$ with probability $4\epsilon$. From the point of view of the white player (Player 2), information set C guarantees an expected utility of $-1 \cdot 1/2 + 2 \cdot 1/2 = 1/2$, while information set D guarantees and expected utility of $0$. So, it is rational for the white player to put as much probability mass as allowed by the lower bounds to action $r$. This is achieved when action $r$ is selected with probability $1 - \epsilon$, and action $s$ with probability $\epsilon$.

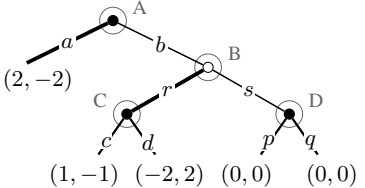

| Action | Probability |
|---|---|
| $a$ | $1 - 4\epsilon$ |
| $b$ | $4\epsilon$ |
| $c, d, p, q$ | $1/2$ |
| $r$ | $1 - \epsilon$ |
| $s$ | $\epsilon$ |

Figure 4: Small perfect-information game that illustrates that uniform $\epsilon$ lower bounds can induce irrational behavior. Black nodes belong to Player 1, the white node belongs to Player 2.

## C  One-sided QPE and trembling linear program formulation

We report the definition of one-sided $\epsilon$-quasi-perfect strategy profiles and equilibrium below for convenience.

**Definition 4** (One-sided quasi-perfect equilibrium). *We call a strategy profile $(\pi_m, \pi_h)$ a one-sided $\epsilon$-quasi-perfect strategy profile if $\pi_h$ is an $\epsilon$-quasi-perfect best response (Definition 2) to $\pi_m$, and $\pi_m$ is a best response to $\pi_h$. We say that $(\pi_m, \pi_h)$ is an one-sided quasi-perfect equilibrium if it is the limit point of one-sided $\epsilon$-quasi-perfect strategy profiles, as $\epsilon \to 0^+$.*

**Theorem 1.** *Every one-sided quasi-perfect equilibrium is a Nash equilibrium.*

*Proof.* The proof is based on a simple continuity argument. Let $(\boldsymbol{x}_m^*, \boldsymbol{x}_h^*)$ be a one-sided QPE. By definition, there exists a sequence $(\boldsymbol{x}_m^t, \boldsymbol{x}_h^t)$ of one-sided $\epsilon^t$-quasi-perfect strategy profiles, such that $\epsilon^t \to 0^+$ and $\boldsymbol{x}_m^t \to \boldsymbol{x}_m^*, \boldsymbol{x}_h^t \to \boldsymbol{x}_h^*$. Because the expected utility of either player is a continuous function of the strategies, it is trivial to show that $\boldsymbol{x}_m^*$ is a best response to $\boldsymbol{x}_h^*$. Indeed, suppose not for the sake of contradiction. Then, there exists $\tilde{\boldsymbol{x}}_m$ such that $(\boldsymbol{x}_m^*)^\top \boldsymbol{A}_m \boldsymbol{x}_h^* < \tilde{\boldsymbol{x}}_m^\top \boldsymbol{A}_m \boldsymbol{x}_h^*$. But since $\boldsymbol{x}_m^t \to \boldsymbol{x}_m^*$ and $\boldsymbol{x}_h^t \to \boldsymbol{x}_h^*$, the inequality must hold when $\boldsymbol{x}_m^*$ and $\boldsymbol{x}_h^*$ are substituted with

$x_m^t$ and $x_h^t$, respectively, provided $t$ is large enough. So, eventually $(x_m^t)^\top A_m x_h^t < \tilde{x}_m^\top A_m x_h^t$, contradicting the hypothesis that $x_m^t$ is a best response to $x_h^t$ for all $t$. So, we are only left with the task of showing that $x_h^*$ is a best response to $x_m^*$ as well. The idea of the proof is similar, but made only slightly more difficult due to the presence of purifications. Suppose once again for the sake of contradiction that $x_h^*$ is not a best response to $x_m^*$. Then, there must exist an information set $I \in \mathcal{I}_h$ reached with positive probability where a strictly suboptimal action $a$ is selected, say with probability $2\delta > 0$. By continuity, for $t$ large enough $I$ is still reached with positive probability, and action $a$ is still strictly suboptimal and selected with probability at least $\delta > 0$. But then, when $t$ is large enough that $\epsilon^t < \delta$, one can extract an $I$-local purification that is $\epsilon^t$-consistent with $x_h^t$ and contains the strictly suboptimal action $a$. Such a purification cannot be optimal, contradicting the hypothesis that $x_h^t$ is an $\epsilon$-quasi-perfect best response to $x_m^t$ at all $t$. □

**Lemma 1.** *Consider the bilinear saddle point problem*

$$\max_{\substack{F_m x_m = f_m \\ x_m \geq 0}} \min_{\substack{F_h x_h = f_h \\ x_h \geq \ell_h(\epsilon)}} x_m^\top A_m x_h \tag{3}$$

*where $\ell_h(\epsilon)$ is as in Equation (1). Then, for any $\epsilon > 0$ for which the domain of the minimization problem is nonempty, any solution to (3) is a one-sided $\epsilon$-quasi-perfect strategy profile.*

*Proof.* Let $(x_m, x_h)$ be a solution to (3). It is evident that $x_m$ is a best response to $x_h$. So, the difficulty in the proof is in showing that $x_h$ is an $\epsilon$-quasi-perfect best response to $x_m$. We do so by only minimally adapting the argument in the proof of Lemma 1 in the original work on QPE by Miltersen and Sørensen [22]. We report the argument with our notation for convenience, striving to maintain a $1 : 1$ relationship with the original proof whenever possible. Let $(x_m, x_h)$ be the solution to (3). Let $I \in \mathcal{I}_h$ be arbitrary, let $x_h'$ be an $I$-local purification of $x_h$, $\epsilon$-consistent with $x_h$, and let $x_h^*$ be an arbitrary $I$-local purification of $x_h$. We will show that $x_h'$ is an optimal $I$-local purification by showing that $x_m^\top A_m x_h^* \geq x_m^\top A_m x_h'$ (note that the payoff matrix is for the machine player, and therefore the human player is minimizing the objective, not maximizing).

We claim that there exists a scalar $\delta > 0$ such that $\tilde{x}_h := x_h + \delta(x_h^* - x_h')$ is a valid sequence form strategy, and that it satisfies $\tilde{x}_h \geq \ell_h(\epsilon)$ (that is, $\tilde{x}_h$ is feasible for the internal minimization problem of (3)). Clearly, $F_h \tilde{x}_h = f_h$ is satisfied (by linearity), so $\tilde{x}_h$ satisfies the sequence-form constraints, and we only have to worry about showing that $\tilde{x}_h \geq \ell_h(\epsilon)$. We will check that condition component-wise, that is, sequence by sequence. Note that by definition of $I$-local purification, the strategies $x_h^*$ and $x_h'$ are identical on all sequences, except potentially on sequences that pass through an action at $I$, so we only have to check these. Furthermore, among these, we only have to worry about the ones to which $x_h'$ assigns non-zero weight. But since $x_h'$ is $\epsilon$-consistent with $x_h$, a trivial induction reveals that the realization weight given by $x_h$ to each of these sequences is *strictly* bigger than $\epsilon^{|\sigma|}$. Hence, the claim follows for some sufficiently small $\delta > 0$. Fix such a $\delta$. Then,

$$x_m^\top A \tilde{x}_h = x_m^\top A_m \left(x_h + \delta(x_h^* - x_h')\right) = x_m^\top A_m x_h + \delta\left(x_m^\top A_m x_h^* - x_m^\top A_m x_h'\right). \tag{4}$$

Now, since $\tilde{x}_h$ is a feasible point for the minimization domain of (3), and since $(x_m, x_h$ is optimal for (3), it must be $x_m^\top A_m x_h \leq x_m^\top A_m \tilde{x}_h$. Plugging the previous inequality into (4) yields $x_m^\top A_m x_h^* \geq x_m^\top A_m x_h'$ as we wanted to show. □

**Proposition 1.** *Any limit point of solutions to the trembling linear program*

$$\mathcal{P}(\epsilon) := \begin{cases} \arg\max_{x_m} & (A_m \ell_h(\epsilon))^\top x_m + (f_h - F_h \ell_h(\epsilon))^\top v \\ \text{s.t.} & ① \; A_m^\top x_m - F_h v \geq 0 \\ & ② \; F_m x_m = f_m \\ & ③ \; x_m \geq 0, \; v \text{ free.} \end{cases}$$

*as the trembling magnitude $\epsilon \to 0^+$ is a one-sided quasi-perfect equilibrium strategy for the machine player.*

*Proof.* We start by dualizing the minimization problem inside of (3), and obtain

$$\min_{\substack{F_h x_h = f_h \\ x_h \geq \ell_h(\epsilon)}} x_m^\top A_m x_h = \begin{cases} \max & f_h^\top x_h + \ell_h(\epsilon)^\top w \\ \text{s.t.} & ① \; F_h^\top v + w = A_m^\top x_m \\ & ② \; w \geq 0, \; v \text{ free.} \end{cases}$$

Next, we eliminate the constraint ① by replacing all occurrences of $\boldsymbol{w}$ with $\boldsymbol{A}_m^\top \boldsymbol{x}_m - \boldsymbol{F}_h^\top \boldsymbol{v}$, thus obtaining the equivalent optimization problem

$$\min_{\substack{\boldsymbol{F}_h \boldsymbol{x}_h = \boldsymbol{f}_h \\ \boldsymbol{x}_h \geq \boldsymbol{\ell}_h(\epsilon)}} \boldsymbol{x}_m^\top \boldsymbol{A}_m \boldsymbol{x}_h = \begin{cases} \max \ (\boldsymbol{A}_m \boldsymbol{\ell}_h(\epsilon))^\top \boldsymbol{x}_m + (\boldsymbol{f}_h - \boldsymbol{F}_h \boldsymbol{\ell}_h(\epsilon))^\top \boldsymbol{v} \\ \text{s.t.} \ ① \ \boldsymbol{A}_m^\top \boldsymbol{x}_m - \boldsymbol{F}_h^\top \boldsymbol{v} \geq \boldsymbol{0} \\ \quad\quad ② \ \boldsymbol{v} \ \text{free.} \end{cases}$$

Finally, we plug the dualized inner minimization problem back into the outer maximization problem of (3), obtaining the statement. □

# D  Additional details about the experiments

## D.1  Description of game instances

**Leduc poker** is a standard benchmark in the extensive-form game-solving community [29]. The game is played with a deck of $R$ unique cards (number of ranks), each of which appears exactly twice in the deck. The game is composed of two rounds. In the first round, each player places an ante of 1 in the pot and is dealt a single private card. A round of betting then takes place, with Player 1 acting first. At most two bets are allowed per player. Then, a card is is revealed face up and another round of betting takes place, with the same dynamics described above. After the two betting round, if one of the players has a pair with the public card, that player wins the pot. Otherwise, the player with the higher card wins the pot. All bets in the first round are worth 1, while all bets in the second round are 2.

**Goofspiel** is another popular benchmark game, originally proposed by Ross [27]. It is a two-player card game, employing three identical decks of $k$ cards each whose values range from 1 to $k$ (in our experiments, $k = 4$). At the beginning of the game, each player gets dealt a full deck as their hand, and the third deck (the "prize" deck) is shuffled and put face down on the board. In each turn, the topmost card from the prize deck is revealed. Then, each player privately picks a card from their hand. This card acts as a bid to win the card that was just revealed from the prize deck. The selected cards are simultaneously revealed, and the highest one wins the prize card. If the players' played cards are equal, the prize card is split. In the experiments, we use the imperfect-information variant of Goofspiel, which has been used multiple times in the literature (*e.g.*, [18]): the players are only informed of who wins each prize, but not of the bid of the opponent.

**River Endgame** The river endgame is structured and parameterized as follows. The game is parameterized by the conditional distribution over hands for each player, current pot size, board state (5 cards dealt to the board), and a betting abstraction. First, Chance deals out hands to the two players according to the conditional hand distribution. We align with Brown and Sandholm [3], and used a simple action abstraction: initial bets are half-pot, full-pot, and all-in, and subsequent raises are full-pot and all-in. The game ends whenever a player folds (the other player wins all money in the pot), calls (a showdown occurs), or both players check as their first action of the game (a showdown occurs). In a showdown the player with the better hands wins the pot. The pot is split in case of a tie.

**Liar's dice** is another standard benchmark in the EFG-solving community [19]. In our instantiation, each of the two players initially privately rolls an unbiased 6-face die. The first player begins bidding, announcing any face value up to 6 and the minimum number of dice that the player believes are showing that value among the dice of both players. Then, each player has two choices during their turn: to make a higher bid, or to challenge the previous bid by declaring the previous bidder a "liar". A bid is higher than the previous one if either the face value is higher, or the number of dice is higher. If the current player challenges the previous bid, all dice are revealed. If the bid is valid, the last bidder wins and obtains a reward of $+1$ while the challenger obtains a negative payoff of $-1$. Otherwise, the challenger wins and gets reward $+1$, and the last bidder obtains reward of $-1$.

## D.2 One-sided QPEs in games with sparsified payoff matrices

As we discuss in Section 5, a key step in our algorithm for computing a one-sided quasi-perfect equilibrium relies is to be able to solve the linear program $\mathcal{P}(\epsilon)$ defined in Proposition 1 for different numerical instantiations of the value of $\epsilon > 0$. Since the solution of the linear programs is the bottleneck of our algorithm, generally speaking the sparsest the formulation of the linear programs $\mathcal{P}(\epsilon)$, the better. The use of *sparsified* payoff matrices was recently shown to help speed up the solution of linear programs representing Nash equilibrium computations [33]. A *sparsification* of the payoff matrix $\boldsymbol{A}_m$ of the machine player is a decomposition of the form $\boldsymbol{A}_m = \hat{\boldsymbol{A}}_m + \boldsymbol{U}_m \boldsymbol{V}_m^\top$, such that the combined number of nonzeros in $\hat{\boldsymbol{A}}_m, \boldsymbol{U}_m$, and $\boldsymbol{V}_m$ is significantly smaller than the number of nonzeros in $\boldsymbol{A}_m$. We now show that any such sparsification can be used in the context of Proposition 1 to improve the sparsity of the constraint matrix. Specifically, we have the following immediate corollary of Proposition 1.

**Proposition 2.** *Let the payoff matrix $\boldsymbol{A}_m$ of the machine player be expressed in sparsified form as*

$$\boldsymbol{A}_m = \hat{\boldsymbol{A}}_m + \boldsymbol{U}_m \boldsymbol{V}_m^\top$$

*for some matrices $\hat{\boldsymbol{A}}_m, \boldsymbol{U}_m, \boldsymbol{V}_m$. Then, any limit point of solutions to the trembling linear program*

$$\mathcal{P}_s(\epsilon) := \begin{cases} \underset{\boldsymbol{x}_m}{\arg\max} \;\; (\boldsymbol{A}_m \boldsymbol{\ell}_h(\epsilon))^\top \boldsymbol{x}_m + (\boldsymbol{f}_h - \boldsymbol{F}_h \boldsymbol{\ell}_h(\epsilon))^\top \boldsymbol{v} \\ \quad \text{s.t.} \;\; \textcircled{1} \;\; \boldsymbol{U}_m \boldsymbol{x}_m - \quad\quad \boldsymbol{y}_m \quad\quad\quad = \boldsymbol{0} \\ \quad\quad\quad \textcircled{2} \;\; \hat{\boldsymbol{A}}_m^\top \boldsymbol{x}_m + \boldsymbol{V}_m \boldsymbol{y}_m - \boldsymbol{F}_h \boldsymbol{v} \geq \boldsymbol{0} \\ \quad\quad\quad \textcircled{3} \;\; \boldsymbol{F}_m \boldsymbol{x}_m \quad\quad\quad\quad\quad = \boldsymbol{f}_m \\ \quad\quad\quad \textcircled{4} \;\; \boldsymbol{x}_m \geq \boldsymbol{0}, \;\; \boldsymbol{y}_m \; \text{free}, \;\; \boldsymbol{v} \; \text{free}. \end{cases}$$

*as the trembling magnitude $\epsilon \to 0^+$ is a one-sided quasi-perfect strategy for the machine player.*

When $\hat{\boldsymbol{A}}_m = \boldsymbol{A}_m$ (that is, no sparsification is computed), constraint $\textcircled{1}$ and variable vector $\boldsymbol{y}_m$ are both empty, thereby reducing $\mathcal{P}_s$ to $\mathcal{P}$.

All remarks in Section 5 about which exhibit a dependence on the trembling magnitude $\epsilon$, apply without changes to the sparsified case as well.

We use the sparsification technique of Zhang and Sandholm [33] to be able to scale to the river endgame.

## D.3 Additional experimental results

Figure 5 shows the game-theoretic performance of our refined one-sided QPE strategies compared to unrefined strategies computed by Gurobi and CPLEX in the two smallest games used in Section 6, which we had to omit from the body of the paper for space reasons. It complements Figure 3. The empirical observations are in line with what was noted in Figure 3.

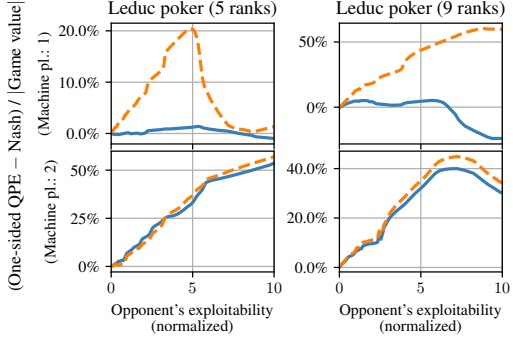

Figure 5: Increase in expected utility achieved by (refined) one-sided QPE strategies compared to the (unrefined) Nash equilibrium strategy for the game computed by Gurobi (solid blue line) and CPLEX (dashed orange line).