# OpenReview forum: "Equilibrium Refinement for the Age of Machines: The One-Sided Quasi-Perfect Equilibrium"
_NeurIPS.cc/2021/Conference — NeurIPS 2021 Poster_

### Official Review · Reviewer_gGTR · 2021-07-12

**Rating:** 8
**Confidence:** 4

**Summary:**

In zero-sum games, playing any Nash equilibrium guarantees optimal performance against an optimal opponent.  However, if the opponent plays suboptimally, playing according to Nash equilibrium may not take full advantage of their mistakes; this is because Nash equilibrium does not require optimal behavior at states that are not reached by mutually optimal play.  Multiple "trembling hand" refinements exist that require optimal play at all states, even those that can be reached only if one or the other of the players play suboptimally.  However, these refinements are typically the limit of equilibria of perturbed games in which each action is played with positive probability, as this probability is sent to 0; this can make them expensive to compute.

This paper considers a special case of the quasi-perfect equilibrium refinement, which it calls the one-sides quasi-perfect equilibrium.  In this refinement, only one agent (the "human") is assumed to tremble; the other agent (the "machine") is assumed to always play optimally; thus, the machine player does not need to account for the possibility that they themselves will err, only that their opponent will err.  The paper shows that it is possible to under these assumptions to efficiently compute a refined equilibrium strategy for the machine player, and demonstrate empirically that a refined strategy will perform substantially better against a suboptimal player (where suboptimal players are taken by intermediate steps of a CFR self-play training run).


**Limitations And Societal Impact:**

The authors address the technical limitations of their work quite carefully.

Regarding societal impact, it would probably be wise to engage a bit more with the fact that the whole premise of the paper is exploiting suboptimal opponents.  In poker this is a morally uncomplicated goal, but if the authors envision this algorithm being used outside the domain of literal games, it becomes more problematic.

**Main Review:**

Overall, I'm very enthusiastic about this paper.

The problem that the paper studies is a long-standing issue in computational game theory.  The one-sided solution is novel (and very clever!), and the paper does an excellent job of giving an intuition for why it enables faster computation.  The empirical results are taken with respect to relatively toy domains, but I find them very convincing evidence that refinements really matter in zero-sum domains.

The paper is very clearly written and well-organized.  The minor exception is the introductory material in section 2, which I found somewhat rushed; if I were not already familiar with sequence-form strategies for imperfect information games, I'm not sure I would have been able to figure it out from this section.  I have a couple of specific suggestions below, but really the whole section could stand to be edited for clarity.

* p.2: "the sequence of actions of Player i on the path from the root to any node in I passes is a superset of the set of actions of the same player on the path from the root to any node in I'.": This is very confusingly written.  Once you have defined sequences it might be simpler to say that any sequence in I' contains an action at I.

* p.3: "containing the probability of all of the player's actions in that sequence": This is unclear; perhaps the product of the probabilities of the player's actions in that sequence?

**Time Spent Reviewing:**

3

---

> ### Author Response · Authors · 2021-08-10
> **Response to reviewer 3**
>
> Thank you for your review and your useful suggestions for improving the writing! We will apply them.

---

### Official Review · Reviewer_S3wD · 2021-07-14

**Rating:** 5
**Confidence:** 5

**Summary:**

The paper introduces a novel refinement of Nash equilibrium for the case where one player is rational and the other may make mistakes. It is a one-sided variant of quasi-perfect equilibrium (QPE). Basic properties, such as existence and correctness of its definition are proven.  The paper further argues that this refinement is simpler to compute than the full QPE and shows how the recent advances in solving trembling LPs can be used to find the solution. In the experimental evaluation, the paper shows that the one-sided QPE can be computed faster than full QPE on a variety of games and that it provides a better value against a suboptimal opponent, in comparison to an arbitrary NE.

**Limitations And Societal Impact:**

The authors nicely addressed some limitations, such as the inability to address non-credible threats, but missed some even more important limitations discussed in the review.  For the societal impact, the goal of the paper is to create strategies that are more effective against humans in zero-sum games, so there might have been some discussion on whether it is a good idea. However, I personally did not miss it there.

**Main Review:**

Originality: The concept of one-sided equilibrium refinements is novel. However, there has been quite some work on one-sided solution concepts in general, where only one of the players is assumed to be irrational, which is not mentioned. Also, the solution technique is a quite straightforward simplification of the solution technique for the full QPE.

Clarity: The paper is very well written, with intuitive examples and clear language. It was a pleasure to read.

Quality: The technical claims in the paper are sound and clear. However, some of the more general terms are overstated. I have two main concerns:

(1) Suitability against humans. I completely agree with the paper in that it is important to study solution concepts where one of the players is rational and the other is not and in particular, if she is a human. However, the paper fails to argue why an equilibrium refinement is desirable in this context and why just not to model the opponent’s irrationality based on quantal response, prospect theory, or a similar model of human decision making. I am not aware of any behavioral economics literature that would suggest that humans play equilibrium refinements. A much more careful positioning w.r.t. computational game theory works taking this approach should be provided (e.g., [A] and the many follow-ups).

(2) Undominated equilibria.  The paper completely ignores the NE refinement of undominated equilibrium, which is much easier to compute than one-sided QPE, has been shown to perform at least as well against imperfect opponents [B], and has been applied in realistic poker endgames [C]. This equilibrium also assumes that one of the players is rational and should have been used as a baseline in the experiments.

Significance: I think the significance of the paper is limited. If I could use LP solving techniques against a human opponent, I would probably not use a NE refinement. If I really wanted to do it, even after reading this paper, I would use the undominated equilibrium. Hence the only remaining significance could be in the theory. Since the computational techniques are quite straightforward generalizations of previous techniques for computing QPE, it is not likely to bring much new reusable knowledge. There is definitely some value in the definition, existence and connection to trembling LPs, but that is in my opinion not enough for a NeurIPS publication.

References:

[A] Yang, Rong, Kiekintveld, C., Ordonez, F., Tambe, M., & John, R.  "Improving resource allocation strategy against human adversaries in security games." Twenty-Second International Joint Conference on Artificial Intelligence. 2011.
[B] Čermák, Jiří, Branislav Bošanský, and Viliam Lisý. "Practical performance of refinements of nash equilibria in extensive-form zero-sum games." ECAI 2014. IOS Press, 2014. 201-206.
[C] Ganzfried, Sam, and Tuomas Sandholm. "Improving performance in imperfect-information games with large state and action spaces by solving endgames." Workshops at the twenty-seventh AAAI conference on artificial intelligence. 2013.

Detailed comments:

line 139: the definition is not clear about what happens if more actions have probability over \epsilon

line 146: the notion of “optimality” used here is most likely not defined

line 376: Why is it meaningful to normalize by the game value? It can be shifted arbitrarily without strategically changing the game and it can be 0, which is not even defined.


After the discussion with the authors, I believe that the authors will make a substantial effort to try to mitigate the concerns above. However, since there is no other round of reviews, I would still lean towards not accepting the paper at the moment, with a strong encouragement to resubmit a modified version. On the other hand, I will not further argue against accepting it either and I have reflected it in my score.


**Time Spent Reviewing:**

4

---

> ### Author Response · Authors · 2021-08-10
> **Response to reviewer 2**
>
> Thanks for your work!
>
> -------
>
> **Q: Concern (1): However, the paper fails to argue why an equilibrium refinement is desirable in this context and why just not to model the opponent’s irrationality based on quantal response, prospect theory, or a similar model of human decision making.**
>
> A: As we mentioned in the response to the first reviewer, sequentially-rational Nash equilibria bring benefits over unrefined Nash in a model-free way, that is, without data about the opponent (unlike the directions you mention that try to model an opponent). They fix some intrinsic room for irrationality in the definition of extensive-form Nash equilibrium, instead of trying to exploit the model of the opponent behavior and going off the equilibrium path (such as with quantal response, prospect theory, or more explicit opponent models).
>
> Empirically studying how much that theoretical benefit translates into practical performance against humans would require setting up human experiments, which is itself a massive undertaking (e.g., multiple Science papers have been written on poker AIs against humans) and the results would still be human dependent. We leave that as a possible direction for future research. The analysis in our paper can be viewed as a first smoke test (in fact, the first of its kind, and on games significantly larger than anything prior refinement technology could scale to), but should not be taken as ultimate proof that refinements bring significant advantages compared to unrefined Nash strategies against human opponents, and we were careful to word our conclusions that way. Our primary goal was to help scale up refinement computation technology to even start to enable those further experiments and investigations on such an important topic (the—mostly theoretical—study of sequential rationality and equilibrium refinements occupied the game theory community for decades in the late 1900s).
>
> We will add a citation to [A].
>
>
> ----------
>
> **Q: Concern (2): The paper completely ignores the NE refinement of undominated equilibrium, which is much easier to compute than one-sided QPE, has been shown to perform at least as well against imperfect opponents [B], and has been applied in realistic poker endgames [C]. This equilibrium also assumes that one of the players is rational and should have been used as a baseline in the experiments.**
>
> A: Thank you for bringing up undominated equilibrium, but we would like to bring up the reasons why that is out of the scope of our paper.  Our paper is about a new solution concept akin to sequential rationality but for settings where one player is perfect (a “machine”).
>
> Undominated equilibria are known not to be sequentially rational. For example, Miltersen and Sorensen [1, page 108] illustrate the weakness of undominated Nash equilibria (UNEs) and their inability to capitalize on mistakes compared to QPEs. UNEs do not preclude a player from hoping for a “gift” (mistake) from the opponent. In other words, while restricting to undominated strategies is a step in the right direction, it does not rule out sequentially-irrational behavior for either player. The solution concepts we start from in this paper—QPEs and EFPEs—guarantee sequential rationality and are a standard solution to this problem. Furthermore, as we wrote in our response to the first reviewer, one of the reasons why QPE is considered superior to EFPE in the literature is the fact that QPE strategies are normal-form perfect, and therefore undominated strategies. So, QPEs guarantee undomination (whereas EFPEs might not). In other words, if you want undomination AND sequential rationality at the same time, QPE is the natural solution concept you would be looking for.
>
> We also think that the statement that “UNE has been shown to perform at least as well against imperfect opponents” is at least arguable. The ECAI paper [B] you mention experimented on extremely small artificial poker variants (six cards in the deck and two betting rounds), and the literature, on the other hand, already previously pointed out examples where undomination does not prevent sequential irrationality — which makes (unrefined) undomination strictly worse than QPE in those examples. So, you seem to be making an overly broad, bold claim.
>
> Regarding [C], they did not compare undomination against QPE or one-sided QPE — because the technology for computing the latter two did not exist at the time.
>
> We will add this discussion in the paper, and cite [B] and [C].
>
> In conclusion, our paper is about sequential rationality and wanted to introduce a solution concept for that consideration, which is applicable when one side is fully rational (a “machine”). If one wants both undomination (as you suggest) AND sequential rationality, QPE is the natural solution. Indeed, that is our starting point. Undominated equilibria were, for all the reasons discussed above, beyond the scope of this paper, but we agree that they could be used as additional benchmarks in future work. Traditional sequential rationality and undomination are apples and oranges: it is well known that they are formally incomparable in that neither implies the other and either can be better than the other depending on the opponent and the game.
>
> [1]: Peter Bro Miltersen and Troels Bjerre Sørensen. SODA 2006.  Computing sequential equilibria for two-player games.
>
>
> ----------
>
> **Q: Conclusion paragraph: If I could use LP solving techniques against a human opponent, I would probably not use a NE refinement. If I really wanted to do it, even after reading this paper, I would use the undominated equilibrium.**
>
> A: Trembling-hand equilibrium refinements are a standard solution to the important problem of sequential irrationality. They have been studied in game theory for decades, as well as in AI. They provide a way of fixing undesirable behavior in Nash strategies in a model-free way. When you say “I would probably not use a NE refinement against a human opponent”, we feel like you are implicitly trying to exploit a model of the opponent, which is beyond the scope of our paper. Undominated equilibria are not sequentially rational in general, but QPEs are both sequentially rational and undominated. Many questions remain open about trembling-hand equilibrium refinements, including their practical performance compared to Nash equilibrium in the large. Before those questions can be studied experimentally, scalable technology for computing trembling-hand refinements needs to be available. Our paper goes in that direction, and we were able to scale the computation of trembling-hand refinements in games that were way too large before. Given the importance of the topic and its extensive theoretical exploration in the literature, we felt that it was very exciting to compute refinements and give the first analysis at this scale.

---

> > ### Comment · Reviewer_S3wD · 2021-08-11
> > **Still not convinced**
> >
> > The authors’ response did not convince me that they are taking my concerns seriously and will adapt the paper sufficiently to mitigate them.
> >
> >
> > (1) Suitability against humans:  I really think that neither this paper nor any of the cited references supports that playing an equilibrium refinement against humans would be a better idea than to use some model of human irrationality. This would be fine if the paper did not claim the equilibrium is for playing against humans in the title and did not use machine and human to refer to the players all over the text. I agree that in some circumstances, it may be more important to minimise exploitability even for the cost of minimal exploitation of the opponent, but this is not what the paper argues. In fact, I think the equilibrium may be much more suitable for playing against other AI players, which may range from very bad to vastly superior opponents, that would exploit the agent’s every suboptimality.
> >
> > Citing [A] will not resolve this issue, the title and referencing the potentially subrational opponent as “human” needs to be changed, in my opinion, to avoid being misleading.
> >
> > (2) Undominated equilibrium: I understand that there are some theoretical corner cases where (one-sided) QPE can perform better than UNE. However, the only empirical evaluation on somewhat realistic settings I am aware of shows that the performance is almost universally worse. And as authors admit in their rebuttal, computing the LP needed to compute UNE is 1-2 orders of magnitude faster than their approach. It is true that the evaluation in [B] is on small games, most likely because of the limitation of the ability of computing QPE, which the authors largely resolve in this paper. However, since it would be trivial to test UNE in their LP-based framework and they did not provide any evidence on this in the rebuttal, I have no reason to think that this property does not hold in the larger games as well.
> >
> > On the other hand, the authors convinced me that since there are important fundamental differences between the solution concepts, there is value in providing an algorithm to compute the one-sided QPE more efficiently, e.g., in order to provide comparison with UNE in a particular class of games.  Even if the comparison shows that UNE is better than QPE in practice, I would be happy to increase my score. However, not presenting the comparison makes the reader believe that this algorithm really is the best known tradeoff between safe exploitation and computational efficiency, which it possibly is not.
> >
> >
> > To sum up, I like the paper and I would be happy to recommend accepting it if the focus on “human” is removed and the empirical comparison to UNE is provided. However, since there will not be another round of review anymore, I prefer not to do so. Since it looks like the other reviewers are positive about the paper at the moment, I kindly ask the authors to reconsider my concerns in preparing the final version, if the paper is accepted.

---

> > > ### Author Response · Authors · 2021-08-25
> > > **Response to reviewer 2**
> > >
> > > First of all, we wanted to express our appreciation for your review. Your comments are excellent, and by all means we are taking them seriously, and we will act on them to improve the paper.
> > >
> > > Regarding point (1), we see what you mean. The trembling-hand refinement of Nash equilibrium we propose indeed is applicable in any situation in which an agent that doesn't make mistakes (a "machine") plays against an imperfect agent. We chose the term "human" to refer to that imperfect agent, but you are right in that really what we meant was just any imperfect agent, including those that are not human. In fact, the main feature of our trembling-hand refinement is that one player doesn't make mistakes, and we do not make particular assumptions about the opponent. We are thinking about amending most references to "machine vs human" to drop the human component to avoid possibly misleading readers into assuming that the refinement is specifically intended to play against humans. And we could even change the title to "Equilibrium Refinement for the Age of Machines: The One-Sided Quasi-Perfect Equilibrium" if the reviewers think that is a better title.
> > >
> > > Regarding point (2): we are glad that we could partially resolve the issue. While our paper focused on trembling-hand perfection and sequential rationality, and we were pretty careful not to make any claims of guaranteed performance against other refinements (for example, we never claimed that one-sided QPE is better than QPE, EFPE, or other non-trembling-hand refinements), we agree that it makes sense to be very careful and point out that the discussion as to which is the the "most appropriate" or best refinement (trembling-hand or not) is still ongoing. It might very well be that undominated Nash, despite not guaranteeing sequential rationality, ends up performing better in practice, against certain types of opponents, or even against all opponents in certain games classes. In this paper, we introduced one-sided QPE, showing that its computation can scale to substantially larger games than what was possible before with trembling-hand technology. While we pointed out some possible pitfalls and shortcomings about one-sided QPE, we provided experiments that intended to show – and do show -- that one-sided QPE is better than Nash, and is a solution concept worth taking into consideration. Other refinements (trembling-hand or not) might be even better than one-sided QPE, including possibly undominated Nash. However, we hope you agree that one-sided QPE has merits, and that this paper already does enough.
> > >
> > > In terms of changes to the paper, in the final conference version we will definitely explain how sequential rationality and undomination are different (as we discussed in the response to reviewers), and point out that there is some evidence that in practice, undomination might end up playing close to sequentially-rational at a fraction of the cost. We will also mention as a future direction of research comparing different refinements in large games. For example, one could compare several solution concepts (including QPE, EFPE, undominated Nash, and one-sided QPE) on different types of opponents (for example, those generated by CFR but also bounded-rational players playing quantal-response equilibrium strategies, for example) in as many games of interest as possible. That would extend the prior analysis you pointed out, to games as large as what trembling-hand refinement technology can handle. In our current paper, we have pushed that boundary further, so such experiments will now be possible at scale (for our solution concept).
> > >
> > > Once again, since you wrote that you were not convinced that we were "taking your concerns seriously and will adapt the paper sufficiently to mitigate them", we want to be extremely clear: we are grateful to you and the other reviewers for your quality job with this submission. No matter what happens with the final decision, you have made our paper better, and your input is valuable and taken into account.

---

> > > > ### Comment · Reviewer_S3wD · 2021-08-25
> > > > **I am more convinced the authors with try to address my concerns in the final version now.**
> > > >
> > > > Thank you for the additional response. Now I believe that the you will make a substantial effort to try to mitigate my concerns. However, since there is no other round of reviews, I would still lean towards not accepting the paper at the moment, with a strong encouragement to resubmit a modified version. On the other hand, I will not further argue against accepting it either and I have reflected it in my score.

---

### Official Review · Reviewer_K3jG · 2021-07-16

**Rating:** 8
**Confidence:** 4

**Summary:**

This paper investigates a trembling-hand refinement of Nash equilibrium in sequential games in which one of the players is considered entirely rational (i.e., a machine), while the second player makes mistakes the first player should account for (i.e., a human). The authors discuss two most commonly studied two-sided variants of trembling equilibria in sequential games - extensive-form perfect equilibrium (EFPE) and quasi-perfect equilibrium (QPE) - and argue that QPE is more suitable for introducing the one-sided variant, as it is both more efficiently computable and considered superior to EFPE in the literature. The paper proves that every one-sided QPE is a Nash equilibrium, and it is guaranteed to exist in every two-player zero-sum sequential game with perfect recall. Moreover, it introduces a so-called trembling linear program the one-sided QPE is a limit of, as the trembling magnitude approaches zero. The solution of the trembling linear program is reached via an algorithm described in the previous work [5], which the authors adapt to take advantage of specific properties of the program for one-sided QPE. In the last part of the manuscript, the performance of one-sided QPE strategies is evaluated (i) against the EPFE and regular QPE strategies in terms of computational time, and (ii) against unrefined Nash strategies in terms of expected utility when the opponent adopts an imperfect strategy computed by CFR. The results suggest that the one-sided QPE could be computed 4-5x (8-10x) faster than two-sided QPE (EFPE) and achieve higher expected utilities than unrefined Nash strategies, especially when the opponent is more exploitable.


**Ethical Concerns:**

I do not have any ethical concerns.

**Limitations And Societal Impact:**

Could you mention some more concrete arguments why QPE is considered superior to EFPE besides computability?

**Main Review:**

The prose of the paper is elegant and natural; the authors managed to introduce all the necessary notions comprehensively and with just enough examples. The paper's motivation is clear, and I believe the one-sided QPE is an equilibrium that deserves to be studied. The authors fittingly discuss the related concepts and how their results fit into the existing literature on trembling-hand equilibria. I find the presented theoretical results sound and conceptually interesting, even though they do not seem too difficult to prove. I appreciate the two "words of caution" in the main text that explain how intuition might sometimes fail us when perfection is one-sided. The experimental results are sufficient to show how the theoretical advantage of one-sided QPE of having the trembling magnitude only in the criterion translates into the faster computation. I also have a few minor questions relating primarily to the experimental part.

I wondered how long did cplex or gurobi take to compute the ordinary Nash strategies compared to the trembling-hand concepts? Also, according to the results, the unrefined Nash strategies sometimes achieve even higher expected utility than one-sided QPE. It is understandable that when the opponent is unexploitable, both approaches should perform about the same. However, I fail to find a reason why would some unrefined Nash strategies outperform the one-sided quasi-perfect strategies for exploitable opponents, as seen, e.g., in the Liar's dice when the exploitability is lower than (roughly) 3. Did the authors attempt to explain such findings?

The text mentions that the CFR ran for 10k iterations. How were the opponents of different exploitability generated? Were they sampled from the CFR strategies up to 10k  (e.g., for 3k, 5k, etc.), or somehow differently? And in case the CFR strategies strayed off the equilibrial path where no behavioral strategy may be defined by the solutions computed by cplex or gurobi, which actions did the machine player take?

Lastly, the motivation brought up consistently across the paper is to compute entirely rational strategies against humans. In the experiments, the imperfect human strategies are modeled by a limited number of CFR iterations. I agree that CFR indeed produces imperfect strategies when the number of iterations is low; still, I am unsure if the CFR strategies actually model human behavior well. Were the imperfections CFR produces ever shown to represent human decision-making well?

It seems intuitive that every (Miltersen-Sorensen) QPE should also be a one-sided QPE. Is that correct?

A few possible typos:
Line 230: “... will be central ...”
Line 258: “... the trembling linear program from Proposition 1 ...”
Def 4. We say that (\pi_m, pi_h) is a one-sided quasi-perfect equilibrium if it is the limit point of one-sided \eps-quasi-perfect strategy profiles.
Lemma 1. … any solution to (3) is a one-sided \eps-quasi-perfect strategy profile.

Overall, I am convinced the manuscript represents a quality work that deserves publication.

### After rebuttal

I would like to thank the authors for answering all my questions. My impression of the paper remains unchanged, I believe it is a good work.


**Time Spent Reviewing:**

25

---

> ### Author Response · Authors · 2021-08-10
> **Response to reviewer 1**
>
> Thanks for your review! Your summary is very much on point. Thanks for the list of typos, the many questions and opportunities for elaboration. We will improve the paper accordingly and we summarize the answers below.
>
> -----------
>
> **Q: I wondered how long did cplex or gurobi take to compute the ordinary Nash strategies compared to the trembling-hand concepts?**
>
> A: Thanks for the question, it made us realize that it would be good (and easy for us) to include data on that in the paper. While one-sided QPE reduces the computational expense gap between regular and refined Nash, the computation of ordinary Nash strategies is still 1-2 orders of magnitude faster than the refined strategies. As we also mention in the conclusions, clearly more work needs to be done before refined Nash can be an appealing drop-in replacement to regular Nash strategies in very large games. Still, we are happy that with one-sided QPE we were able to compute an exact refined Nash strategy in a real poker endgame with tens of million of leaves, a task that would have been unthinkable with prior technology.
>
>
> ----------
>
> **Q: I fail to find a reason why would some unrefined Nash strategies outperform the one-sided quasi-perfect strategies for exploitable opponents, as seen, e.g., in the Liar's dice when the exploitability is lower than (roughly) 3. Did the authors attempt to explain such findings?**
>
> A: That’s an excellent observation. We looked into it before submitting the paper, and concluded that it was just a happy coincidence that Gurobi and CPLEX happened to randomly compute a strategy that was more exploitative than one-sided QPE for the specific irrationality of the CFR agents at that level of exploitability. At the end of the day, in principle it is possible (though extremely unlikely) that Gurobi and CPLEX luck out and compute sequentially-rational strategies despite no constraints in that direction being imposed. That’s partly why we thought the type of analysis we do in our paper was important to be had. And while there are some data points like the one you brought up, by far most of the experimental results points in the direction that it is experimentally almost never the case that Gurobi and CPLEX luck out that way.
>
>
> ----------
>
>
> Q: How were the opponents of different exploitability generated? Were they sampled from the CFR strategies up to 10k (e.g., for 3k, 5k, etc.), or somehow differently?
>
> A: After each iteration of CFR, we computed the exploitability of the agent. Then, we sorted all the agents by exploitability. For each agent, we computed the expected utility against the refined one-sided QPE strategy and against the (unrefined) Nash strategies computed by Gurobi and CPLEX.
>
>
> ----------
>
> **Q: I agree that CFR indeed produces imperfect strategies when the number of iterations is low; still, I am unsure if the CFR strategies actually model human behavior well. Were the imperfections CFR produces ever shown to represent human decision-making well?**
>
> A: Excellent point. Unfortunately, the question as to what makes a good model of human decision making is hard to answer in a satisfying way, and it ties into behavioral economics. We also suspect that the answer heavily depends on the audience: in the case of poker endgames, for example, top professionals probably would tend to play very differently than amateurs, rendering modeling human decision making (and mistakes) very context dependent. One appeal of sequentially-rational Nash equilibria though is that (in theory) they bring benefits over unrefined Nash in a model-free way. In other words, they fix some intrinsic room for irrationality in the definition of extensive-form Nash equilibrium, instead of trying to exploit the model of the opponent behavior by going off the equilibrium path. Empirically studying how much that theoretical benefit translates into practical performance against humans would require setting up human experiments, which is itself a massive undertaking (e.g., multiple Science papers have been written on poker AIs against humans) and the results would still be human dependent. We leave that as a possible direction for future research.
>
> The analysis in our paper can be viewed as a first smoke test (in fact, the first of its kind, and on games significantly larger than anything prior refinement technology could scale to), but should not be taken as ultimate proof that refinements bring significant advantages compared to unrefined Nash strategies against human opponents, and we were careful to word our conclusions that way. Our primary goal was to help scale up refinement computation technology to even start to enable those further experiments and investigations on such an important topic (the—mostly theoretical---study of sequential rationality and equilibrium refinements occupied the game theory community for decades in the late 1900s).
>
>
> ----------
>
> **Q: It seems intuitive that every (Miltersen-Sorensen) QPE should also be a one-sided QPE. Is that correct?**
>
> A: That’s an interesting question. We are pretty sure the answer is yes. We have a proof in mind, but it is not fully ready yet. We will double check the proof and include this result in the final version if the proof works.
>
>
> ----------
>
> **Q: Could you mention some more concrete arguments why QPE is considered superior to EFPE besides computability?**
>
> A: Sure! The debate to which refinement is preferable dates back to at least the year 1982, and is fraught with misconceptions and incorrect claims, as Mertens [1] crisply summarized in 1995. We believe that QPE is superior to EFPE for at least the following reasons:
>
> 1. An EFPE may prescribe the players to play weakly dominated strategies, while a QPE never does.
> 2. In two-player games, a QPE is also a perfect equilibrium of the normal form, whereas EFPE is not. This led Mertens to write: “Observe that the "quasi-perfect" equilibria [..] are still sequential--and sequential equilibria have all backward-induction properties (e.g., Kohlberg and Mertens, 1986)--but are at the same time normal form perfect--which can be viewed as the strong version of undominated. (And every proper equilibrium is quasi-perfect.) Thus, by some irony of terminology, the “quasi”-concept seems in fact far superior to the original unqualified perfection itself. And it seems to be the one called for in a number of applications involving, e.g., voting situations, like the definition of durable mechanisms, in order to use a single concept, which is applied to the whole game, rather than an ad hoc mixture of admissibility requirements in the embedded voting situation and of sequentiality requirements in the overall game”.
> 3. As shown by Miltersen and Sorensen, a QPE can be computed as the limit solution to a perturbed LP where the perturbation only appears on the right hand side of the constraints. This renders computing a QPE more convenient than EFPE in practice.
>
> [1] Mertens, 1995. “Two Examples of Strategic Equilibrium“. Games and Economic Behavior.

---

### Decision · Program_Chairs · 2021-09-27

**Decision:**

Accept (Poster)

**Comment:**

The paper presents a refinement of Nash equilibrium for cases where one player is rational and the other is not. Some nice properties have been presented about the one-sided quasi-perfect equilibrium (QPE). The paper is well-written with very valuable insights. Most concerns raised in the reviews have been addressed by the authors. The contents are solid enough to justify for a publication, although the significance of the work highly relies on whether we can find practical scenarios of one-sided QPE.